# Purinergic Signalling in Allogeneic Haematopoietic Stem Cell Transplantation and Graft-versus-Host Disease

**DOI:** 10.3390/ijms22158343

**Published:** 2021-08-03

**Authors:** Peter Cuthbertson, Nicholas J. Geraghty, Sam R. Adhikary, Katrina M. Bird, Stephen J. Fuller, Debbie Watson, Ronald Sluyter

**Affiliations:** 1Illawarra Health and Medical Research Institute, Wollongong, NSW 2522, Australia; pc859@uowmail.edu.au (P.C.); geraghty@uow.edu.au (N.J.G.); Sam.Adhikary@qimrberghofer.edu.au (S.R.A.); kb215@uowmail.edu.au (K.M.B.); 2Molecular Horizons and School of Chemistry and Molecular Bioscience, University of Wollongong, Wollongong, NSW 2522, Australia; 3Sydney Medical School Nepean, University of Sydney, Nepean Hospital, Penrith, NSW 2747, Australia; stephen.fuller@sydney.edu.au

**Keywords:** P2X7 receptor, P2Y receptor, adenosine receptor, CD39, CD73, ATP

## Abstract

Allogeneic haematopoietic stem cell transplantation (allo-HSCT) is a curative therapy for blood cancers and other haematological disorders. However, allo-HSCT leads to graft-versus-host disease (GVHD), a severe and often lethal immunological response, in the majority of transplant recipients. Current therapies for GVHD are limited and often reduce the effectiveness of allo-HSCT. Therefore, pro- and anti-inflammatory factors contributing to disease need to be explored in order to identify new treatment targets. Purinergic signalling plays important roles in haematopoiesis, inflammation and immunity, and recent evidence suggests that it can also affect haematopoietic stem cell transplantation and GVHD development. This review provides a detailed assessment of the emerging roles of purinergic receptors, most notably P2X7, P2Y_2_ and A_2A_ receptors, and ectoenzymes, CD39 and CD73, in GVHD.

## 1. Introduction

Blood cancers are a highly complex group of disorders that arise from the malignant transformation of cells of the blood, bone marrow or lymphatic system. These malignancies account for approximately 10% of all cancers worldwide [1]. Allogeneic haematopoietic stem cell transplantation (allo-HSCT) is a curative therapy for blood cancers and other haematological disorders [2]. Allo-HSCT has dual beneficial effects. First, it restores the immune system of the recipient that has been diminished by myeloablative therapy. Second, donor cells provide anti-tumour immunity known as the graft-versus-tumour (GVT) effect to prevent malignant disease relapse. However, allo-HSCT also causes graft-versus-host disease (GVHD) in the majority of recipients [3]. GVHD is a severe and often lethal immunological response against host tissues, mediated by donor T cells [4,5]. Therapies for GVHD are currently limited to broad range immunosuppression and prophylactic T cell depletion strategies, however cancer relapse is still a major cause of mortality [6]. Therefore, new and better treatment options that limit GVHD but maintain the GVT effect, need to be explored. This requires a deeper understanding of which factors contribute to GVHD following allo-HSCT.

Purinergic signalling is a form of cell-to-cell communication involving extracellular nucleosides and nucleotides, and their respective cell surface receptors, with roles in health and disease [7]. Recent evidence indicates roles for these receptors in both allo-HSCT and GVHD. Thus, molecules of the purinergic signalling pathway are emerging as potential therapeutic targets for preventing GVHD development and progression in allo-HSCT recipients. This review will present a brief overview of allo-HSCT including the GVT effect and GVHD, and purinergic signalling. This review will then provide a detailed assessment of our current understanding of the roles of purinergic receptors and other molecules in GVHD.

## 2. Allogeneic Haematopoietic Stem Cell Transplantation and the Graft-Versus-Tumour Effect

Allo-HSCTs were first performed over 50 years ago [8,9] and this treatment is now a well-established therapeutic option for patients with haematological malignancies [2]. In general, allo-HSCT is used to replace an abnormal non-malignant haematopoietic system with haematopoietic stem cells (HSCs) from a healthy donor, or to enable higher doses of myeloablative therapy to treat malignancy. Allo-HSCT is possible because of the ability of the HSCs to home to the bone marrow and replace the entire haematopoietic system after intravenous injection [10]. Prior to allo-HSCT, the recipient receives myeloablative chemotherapy and/or radiotherapy that is designed to eradicate the underlying disease and immunosuppress the patient to prevent immunologically mediated rejection of the transplanted donor HSCs [11,12]. Bone marrow cells or peripheral blood HSCs collected from the donor are infused into the recipient [13]. The donor HSCs “home” to the site of bone marrow in the recipient and engraft in their new haematopoietic microenvironment [13]. Following myeloablative therapy, peripheral blood cell counts decrease over several days and haematopoietic growth factors are administered to the patient to increase the rate of donor stem cell engraftment [14]. Transplanted stem cells start to appear after a week and blood counts fully recover after three to four weeks [14].

A major therapeutic benefit of allo-HSCT is the GVT effect. GVT was first demonstrated in a murine model where allo-HSCT eradicated residual leukaemia but also caused a fatal “wasting syndrome” characterised by diarrhoea [15], now recognised as GVHD. The GVT effect has since been defined in considerable detail in allogeneic [16,17] and humanised mouse models [18,19,20], and has been translated to human studies [21,22]. It is now widely appreciated that the GVT effect is one of the most effective and clinically successful forms of immunotherapy following allo-HSCT.

Allogeneic mouse models are commonly used to study the GVT effect and involve the transplantation of bone marrow or splenocytes from C57BL/6 mice to BALB/c mice, followed by transplantation of cancer cells such as A20 or BCL1 murine lymphoma cells [16,17]. These models have been used to demonstrate that CD8^+^ T cells are the main drivers of the GVT effect [17]. Humanised mouse models involving the transplantation of human peripheral blood mononuclear cells (PBMCs) and human cancer cells, such as THP-1 acute monocyte leukaemia cells, have been used to examine the role of human immune cells in GVT immunity [19,20].

Despite the life-saving benefits of the GVT effect, T cells that are transfused with the graft can also react against tissues of the genetically different recipient causing GVHD, which affects mainly the skin, liver, and gastrointestinal tract but may also target other organs [23]. The major determinant for alloreactivity and choosing an allogeneic donor is the degree of matching at loci in the major histocompatibility complex (MHC) which includes human leukocyte antigens (HLA) encoded by MHC class I and class II genes. The two major approaches to preventing acute GVHD are pharmacologic immunosuppression, commenced early after allo-HSCT, and T cell depletion of the donor graft, however despite prophylaxis approximately 30% of recipients of stem cells from matched siblings and up to 50% from unrelated donors will develop acute GVHD, and chronic GVHD is seen in 30–70% of recipients [3,24,25].

## 3. Acute Graft-versus-Host Disease

Acute GVHD, hereafter referred to as GVHD (unless stated otherwise), has a complex pathophysiology, which initially begins with damage to host tissue by the myeloablative therapy used in allo-HSCT. This causes the release of damage-associated molecular patterns, including adenosine 5′-triphosphate (ATP), which promote the maturation of host dendritic cells (DCs) through up-regulation of MHC class I and class II molecules [26]. Host DCs then activate donor T cells in the graft, which migrate to GVHD target organs and release pro-inflammatory cytokines, including interferon (IFN)-γ [27] and interleukin (IL)-17 which drive inflammatory damage [28]. Although donor CD4^+^ and CD8^+^ T cells are the main mediators of GVHD [4,5], there are several other immune cell subsets involved in its pathogenesis. Neutrophils are associated with intestinal GVHD [29,30] and macrophages contribute to acute and chronic GVHD pathogenesis [31]. Conversely regulatory T cells (Tregs) are important for preventing GVHD development in allo-HSCT recipients [32,33]. Natural killer (NK) cells can have regulatory [34,35] or pathogenic [36,37] roles in GVHD, but are also important for mediating the GVT effect [38]. As a result of this complex pathophysiology current treatments have limited success. Therefore, pre-clinical GVHD models are important tools for identifying novel therapeutics.

As for the GVT studies above, allogeneic mouse models have been the main method used to investigate GVHD, providing vast insight into the pathophysiology of the disease [39]. These mouse models of GVHD typically involve transferring T cell depleted bone marrow, with splenocytes (as a source of T cells) or purified T cells, from one mouse strain (donor) into a second mouse strain (recipient) that has been lethally irradiated. These models have been integral in identifying the role of immune cells and cellular signalling pathways involved in GVHD and testing of therapeutic agents. However, these therapies have not always translated to the clinic due to the inability of GVHD mouse models to fully replicate human immune responses.

This limitation is partially overcome with the use of humanised mouse models of GVHD, which involve the transfer of human immune cells, or tissue, into immunodeficient mice [39]. This immunodeficiency allows the engraftment of human immune cells, and the development of a functional human immune system in these mice. Two mouse strains commonly used for humanised models of GVHD are NOD.Cg-*Prkdc^scid^Il2rg^null^* (NSG) and NOD.Cg-*Prkdc^scid^ Il2rg^tm1Sug^* (NOG). These strains are near-identical, except the former encodes a complete null mutation in the *Il2rg* gene and the latter an *Il2rg* gene mutation giving rise to a non-functional IL-2 receptor [40]. These mice are typically injected with human PBMCs, initiating CD4^+^ and CD8^+^ T cell-mediated GVHD [41,42]. GVHD in humanised mice mediates damage to the same tissues as observed in humans, namely the skin, liver, gastrointestinal tract and lungs [42,43,44,45]. These models are useful for pre-clinical investigations of novel therapeutic strategies for GVHD such as the adoptive transfer of Tregs [46,47] or T cell depletion strategies [48,49]. However, current treatment options do not always prevent GVHD or cancer relapse, so new therapeutics that retain the GVT effect but prevent GVHD are needed.

## 4. Purinergic Signalling

Purinergic signalling is a form of cell-to-cell communication comprising a network of extracellular nucleotides and nucleosides, cell surface receptors, ecto-enzymes and nucleotide release pathways [50]. Purinergic signalling is important in various physiological systems [7] including the immune system [51]. This network is comprised of P1 receptors activated by extracellular adenosine, and P2 receptors typically activated by extracellular ATP but in some cases by other extracellular nucleotides [50]. Adenosine receptors are G-protein coupled receptors comprising four subtypes (A_1_, A_2A_, A_2B_ and A_3_) [52]. P2 receptors comprise two subclasses: P2X and P2Y. P2X receptors are trimeric ligand-gated ion channels comprising seven subunits (P2X1, P2X2, P2X3, P2X4, P2X5, P2X6 and P2X7), which can assemble as homomeric or heteromeric channels [53]. P2Y receptors are G-protein coupled receptors comprising eight subtypes (P2Y_1_, P2Y_2_, P2Y_4_, P2Y_6_, P2Y_11_, P2Y_12_, P2Y_13_ and P2Y_14_) [54,55].

Cells release nucleotides and nucleosides by specific (hemichannels, transporters, microvesicles and exocytosis) and non-specific pathways (cell stress and cell death) [7]. The availability of extracellular nucleotides is further controlled by cell-surface adenylate kinase and nucleoside diphosphate kinases, which convert extracellular adenosine 5′-monophosphate (AMP) to adenosine 5′-diphosphate (ADP) and ATP, as well as a cell-surface ATP synthase [56,57]. The amount of extracellular nucleotides and nucleosides can also be regulated by ecto-nucleotidases, which hydrolyse nucleotides and nucleosides, and comprise the ecto-nucleoside triphosphate diphosphohydrolase (E-NTPDase), ecto-5′-nucleotidase, ecto-nucleotide pyrophosphatase/phosphodiesterase, and alkaline phosphatase families [56,57]. Of these enzymes, ecto-nucleoside triphosphate diphosphohydrolase 1 (E-NTPDase1/CD39), which hydrolyses ATP/ADP to AMP, and ecto-5′-nucleotidase (CD73), which hydrolyses AMP to adenosine, play central roles in inflammation and immunity [58].

## 5. Purinergic Signalling in Allogeneic Haematopoietic Stem Cell Transplantation

This review focuses on the role of purinergic signalling in GVHD including the use of pharmacological agents and genetic approaches to explore this role. However, purinergic signalling also has multiple roles in allo-HSCT, which will be discussed briefly first. Purinergic signalling modulates haematopoiesis and HSCs impacting cell homeostasis, differentiation, mobilisation and death (reviewed by [59,60,61]). P2X and P2Y receptors are found in all haematopoietic cell types, including lymphoid and myeloid cells [61]. Adenosine receptors, and CD39 and CD73 are also found on a range of haematopoietic cells but most prominently on immune cells [58,62]. While HSCs reside in the bone marrow niche, there are also circulating HSCs that play an important role in sensing danger signals from inflammatory damage or injury [61]. Collectively, this evidence supports a role for purinergic signalling in allo-HSCT.

A link between extracellular nucleotides such as ATP and mobilisation of HSCs has been shown with the P2 receptors and NLRP3 inflammasome being involved in the mobilisation of HSCs to the peripheral blood [63]. In particular, ATP and P2X7 play a role in haematopoiesis and mobilisation of HSCs, and trafficking of granulocytes and monocytes [60]. Supporting this, HSCs with single nucleotide polymorphisms (SNPs) (rs2230912 and rs1718119) in the *P2RX7* gene (Q460R and A348T, respectively) show an increase in CD34^+^ HSC mobilisation [64]. Further, studies have demonstrated that ATP promotes, and adenosine inhibits, the migration of HSCs [65,66]. Uridine 5′-triphosphate (UTP) stimulation of P2Y receptors can also increase the migration of HSCs [67] and P2Y_14_ has been shown to modulate HSCs under stressful conditions [68]. CD39 and CD73 are also highly expressed on HSCs [69]. Adenosine generated by CD39 on CD150^high^ Tregs in the bone marrow niche can control HSC quiescence, and both CD150^high^ conventional CD4^+^ T cells and Tregs can modulate HSCs via CD73 [70]. CD73 knockout mice show increased HSC mobilisation compared to wild-type (*wt*) mice [65]. Therefore, CD39 and CD73 prevent ATP-mediated differentiation of HSCs by degrading ATP to adenosine, a negative feedback loop, which leads to maintaining HSCs in a quiescent state via A_2A_ activation [65,70].

Purinergic signalling also plays an important role in allo-HSCT. Prior to transplant, granulocyte-colony stimulating factor (G-CSF) mobilisation of HSCs to peripheral blood makes HSCs easily accessible and improves engraftment [71]. However, some individuals do not mobilise adequate numbers of HSCs and strategies to generate more efficient protocols are required [60]. In humans, lower P2X7 expression is associated with lower HSC mobilisation following G-CSF [64]. Pre-conditioning regimes, including chemotherapy and radiotherapy, prior to allo-HSCT may also result in tissue damage and release of danger signals, including ATP, that activate purinergic signalling pathways [72,73].

ATP and its analogues can impact HSC differentiation and engraftment, and play a role in trafficking of HSCs to bone marrow niches following allo-HSCT [60]. Pannexin-1-mediated release of ATP is also important for HSC trafficking, with pannexin-1 blockade leading to defects in HSC homing and engraftment in mice [74]. Therefore, targeting pathways associated with purinergic signalling including the NLRP3 inflammasome, pannexin-1 or P2X7 to reduce inflammatory immune responses or targeting CD39 or CD73 that hydrolyse extracellular ATP to adenosine to promote immunosuppression may improve HSC engraftment and allo-HSCT outcomes.

Identifying SNPs in genes encoding the NLRP3 inflammasome complex, P2X receptors, CD39 (*ENTPD1*) or CD73 (*NT5E*) may establish important predictive or prognostic markers for allo-HSCT outcomes. Donor and recipient SNP variants in the *NLRP3* gene have been examined in HLA-identical sibling allo-HSCT, with only the donor rs10925027 intronic TT genotype being associated with disease relapse [75]. Studies of *P2RX7* SNPs in both donor and recipients of allo-HSCT have shown limited effects on allo-HSCT outcomes. A small cohort showed that the loss-of-function (LOF) rs3751143 *P2RX7* SNP (E496A) in homozygous doses was associated with poorer overall survival [76], but this was not replicated in a larger study [77]. More recently a study examined 16 *P2RX7* SNPs in allo-HSCT donors and recipients [78]. Although limited by sample size, this study revealed that the missense rs3751143 *P2RX7* SNP in homozygous dosage may be associated with overall survival, while the rare LOF rs1653624 *P2RX7* SNP (I568N) may be associated with reduced survival due to infection-related deaths. These studies indicate that *P2RX7* (P2X7) SNPs are unlikely to be biomarkers or predictors of allo-HSCT outcome. To the best of our knowledge there are no reports examining human donor and recipient *ENTPD1* (CD39), *NT5E* (CD73) and *ADORA2A* (A_2A_) genotypes and allo-HSCT outcomes in the clinical setting. Finally, it should be noted that purinergic receptors and molecules play emerging roles in blood cancer cells [79,80], but this is beyond the scope of this review.

## 6. Purinergic Signalling in GVHD

Purinergic signalling has an important role in the progression of GVHD which has been largely elucidated through the use of pharmacological agents as well as knockout mice. The purinergic molecules often highlighted as playing key roles in inflammation and immunity, namely A_2A_, P2X7, P2Y_2_, CD39 and CD73 [81], are also those associated with the promotion or prevention of GVHD [26,82]. The distribution and roles of these purinergic molecules in the immune system were recently reviewed [81,83]. In general, these molecules can combine to form a hypothetical pathway of cellular activation and regulation within the immune system (Figure 1). ATP released during inflammation and cell damage can stimulate P2X7 on leukocytes to drive various inflammatory events such as DC migration [84], pro-inflammatory cytokine release from macrophages and DCs [85], and effector T cell activation [86]. Additionally, UTP can be released along with ATP to activate P2Y_2_ to further promote leukocyte migration [87,88] and cytokine release [89,90]. ATP can be sequentially degraded by CD39 and CD73 to yield adenosine, thereby activating A_2A_ on leukocytes and resulting in the impairment of inflammatory events such as reduced pro-inflammatory cytokine release from T cells [91,92].

A number of compounds have been studied that inhibit purinergic signalling molecules [81,83], however interpretation of the effects of these antagonists in both in vitro and in vivo settings is not straightforward. For example, the efficacy of A_2A_ and P2X7 antagonists differs between species [93,94], which is relevant when studying humanised mouse models of GVHD. Furthermore, it should be noted that inhibitors such as caffeine are non-selective adenosine receptor antagonists, impairing the activity of all four subtypes [93]. Similarly, pyridoxalphosphate-6-azophenyl-2’,4’-disulfonic acid (PPADS) and Brilliant Blue G (BBG) are non-selective P2X7 antagonists that inhibit the majority of P2X receptors and, in the case of PPADS, P2Y_1_ and P2Y_2_ [95]. Another complicating factor is that both A-438079 and BBG impair the ATP-release channel, pannexin-1 [96], and this may impair P2X7-mediated responses by blocking both P2X7 activation and reducing extracellular ATP. KN-62 was originally developed as a calmodulin-dependent protein kinase II antagonist [97] but was subsequently shown to potently inhibit P2X7 [98]. Stavudine is a nucleoside reverse transcriptase inhibitor [99] that impairs P2X7 pore formation but not channel activity [100]. However, a comprehensive pharmacological characterisation of this compound against P2X7 is lacking. Of note, a second nucleoside reverse transcriptase inhibitor, azidothymidine (Zidovudine) [99], binds to the allosteric binding site of P2X7 and impairs P2X7-mediated calcium fluxes and dye uptake in myoblasts [101].

In addition, the action and specificity of the ecto-enzyme inhibitors, α,β-methylene ADP (APCP) and polyoxotungstate-1 (POM-1) used in studies of GVHD, have not been completely defined. APCP is commonly regarded as an inhibitor of CD73, with in vitro enzymatic data showing it does not impair CD39 [102]. In contrast, one study has demonstrated that this compound can inhibit human CD39 heterologously expressed in HEK293 cells [103]. It remains to be determined if APCP can directly impair CD39 from other species, however APCP impairment reversed the extended thrombosis time in human CD39 expressing *Nt5e*^−/−^ transgenic mice [103]. APCP can also modulate P2Y receptor activation [104] further complicating its use in vivo. Finally, POM-1, an inhibitor of CD39 and other NTPDases [105] can also block P2X- and P2Y-mediated calcium responses in macrophages, and impair some but not all features of P2X7 activation in these cells [106], again muddling its use in vivo.

A further limitation to past studies of purinergic signalling in GVHD has been the use of the compounds above in the absence of routine clinical GVHD therapies including corticosteroids and other drugs. However, the interaction of these clinical drugs with purinergic pathways involved in GVHD is largely unknown but remains a possibility. In this regard, the corticosteroid prednisone has been shown to reduce both P2X7 in PBMCs from a patient with the inflammatory disorder Schnitzler’s syndrome [107] and P2X3 in the joints of rats with inflammatory arthritis [108]. Furthermore, a study of asthma patients revealed that those with low P2X7 activity were less likely to benefit from inhaled corticosteroids [109]. Thus, a potential relationship may also exist between purinergic pathways and corticosteroid therapy failure in patients with GVHD.

## 7. P2X7 Receptor Signalling in GVHD

There is increasing evidence that ATP and P2X7 play important roles in GVHD development. This evidence comes from studies examining the effects of ATP degradation, P2X7 expression in murine and human GVHD, P2X7 receptor blockade, and comparisons in P2X7 knockout mice or humans with *P2XR7* SNPs. Together the evidence indicates that the ATP/P2X7 signalling axis initiates pro-inflammatory effects by activating host DCs and causing dysfunction and/or destruction of cells with suppressor activity.

Studies of allogeneic and humanised mouse models indicate a role for extracellular ATP in promoting GVHD (Table 1). ATP is increased in the peritoneal fluid of allo-HSCT patients with GVHD compared to allo-HSCT recipients without GVHD [72]. ATP is also increased in the peritoneal fluid shortly after pre-conditioning and increased in the gut during disease progression in an allogeneic mouse model of GVHD [72,110]. Apyrase, a soluble ATP diphosphohydrolase (ATPDase), injected during the first week post-transplantation reduced apoptosis and inflammation in target organs and serum IFN-γ, and increased survival in an allogeneic mouse model of GVHD [72,110] indicating a direct role for extracellular ATP in mediating GVHD progression following pre-conditioning and transplantation. Indirect evidence for extracellular ATP in GVHD is also revealed through the pharmacological blockade of the CD39/CD73 pathway, which results in increased or sustained extracellular ATP concentrations. Blockade of CD39/CD73 with APCP (days 0–6 or twice weekly) reduced survival, increased T cell proliferation and pro-inflammatory cytokines, and increased GVT immunity in an allogeneic mouse model [111,112]. Furthermore, treatment with APCP (days 0–6) in a humanised mouse model of GVHD, which does not involve pre-conditioning, resulted in worsened disease with increased weight loss, liver apoptosis and serum human IL-2 [113]. This latter finding supports the notion that ATP is released during GVHD progression and that the initial pre-conditioning regime used in allogeneic mouse studies is not essential for ATP release.

The majority of studies to date indicate that P2X7 is the major P2X receptor involved in GVHD progression following ATP release. Human *P2RX7* expression is increased in the PBMCs of human patients with GVHD, compared to allo-HSCT recipients without GVHD or healthy controls [72]. Whilst mouse *P2rx7* expression is increased in the liver, spleen and thymus of allogeneic mice with GVHD [72,114], and in the duodenum, ileum and skin of humanised mice with GVHD [45]. Moreover, P2X7 protein is increased on antigen presenting cells (APCs) in Peyer’s patches and in the colon of patients with GVHD [72]. This increase in P2X7 on APCs may be dependent on *mir-188* expression, as deficiency of this microRNA results in decreased *P2rx7* in murine DCs, corresponding to decreased GVHD in *mir-188* knockout recipients [115]. Murine *P2rx4* is also increased in the gut and skin of humanised mice with GVHD [45], but a role for P2X4 in this disease remains to be investigated.

Pharmacological blockade of P2X7 reduces GVHD in both allogeneic and humanised mouse models of disease (Table 2). The P2X7 antagonists PPADS (days 0–10), KN62 (days 0–10) or A-438079 (days 0–4) increased survival in an allogeneic mouse model of GVHD [72,110]. PPADS treatment also decreased serum IFN-γ and tissue inflammation, which coincided with increased Tregs [72]. Similarly, PPADS treatment (days 0–10) reduced histological GVHD and increased human Tregs in a humanised mouse model of GVHD [117]. Treatment with stavudine (from days −1–9) increased survival and decreased serum pro-inflammatory cytokines, including IFN-γ, in an allogeneic mouse model of GVHD [100]. However, as noted above, the mechanism by which stavudine inhibits P2X7 remains to be fully established. The P2X7 antagonist BBG (twice weekly for 4 weeks) also reduced weight loss, liver inflammation and inflammatory cytokine expression or release, as well as decreasing P2X7 in allogeneic mice with GVHD [114]. BBG (days 0, 2, 4, 6, 8, 10) also reduced serum IFN-γ and inflammation in the liver, skin and small intestine in a humanised mouse model of GVHD [118]. Furthermore, daily BBG treatment over this same time period (days 0–10) reduced clinical and histological GVHD, and serum IFN-γ, and increased the proportions of human Tregs and B cells in a humanised mouse model of GVHD [117]. However, long-term treatment with BBG (thrice weekly until endpoint) only reduced liver inflammation but not serum IFN-γ [119]. These differences suggest that timing of P2X7 antagonism may alter disease outcomes in GVHD. Regardless, these four studies with BBG suggest this compound is having a consistent effect on reducing liver GVHD, perhaps indicating that this compound accumulates at higher concentrations in the liver than other tissues and may be acting at the site of tissue damage.

Genetic studies in mice confirm that P2X7 is involved in GVHD progression and provide further insight to the potential mechanism of this receptor in this disease. Comparisons of *P2rx7^−/−^* and *wt* mice as either the source of donor cells or recipient mice indicated that host, but not donor, P2X7 contributes to GVHD progression in an allogeneic mouse model [72]. Furthermore, bone marrow chimeras in *P2rx7*^−/−^ or *wt* mice demonstrated that P2X7 in the host haematopoietic system, but not non-haematopoietic tissues, influences GVHD in allogeneic mice [72]. Finally, bone marrow chimeras with a host *P2rx7*^−/−^ haematopoietic system including *P2rx7*^−/−^ DCs had reduced disease compared to mice with a host *P2rx7*^−/−^ haematopoietic system but *wt* DCs, indicating that P2X7 on host DCs is contributing to GVHD development [72].

The importance of host P2X7 in GVHD is supported by studies of human *P2RX7* SNPs in mice and humans. NSG mice injected with PBMCs from donors with either a LOF or gain-of-function (GOF) *P2RX7* haplotype display similar GVHD development [120]. Likewise, donor *P2RX7* genotype did not influence rates or severity of GVHD in allo-HSCT recipients [78]. Conversely, the LOF rs3751143 SNP in the host was associated with improved survival, however, other LOF SNPs in the host were not associated with reduced GVHD and the LOF rs1653624 SNP was associated with increased GVHD [78].

The above studies indicate that host P2X7 is important in GVHD development however, they do not exclude the potential contribution of donor P2X7 influencing disease. P2X7 activation can promote the activation and proliferation of CD4^+^ T cells in an IL-2 dependant manner [86] and can help promote the metabolic fitness, maintenance and survival of memory CD8^+^ T cells [121,122], thus the possibility remains that P2X7 may contribute to the activation and survival of donor effector T cells in GVHD. Additionally, P2X7 activation can inhibit the suppressive action and stability of Tregs, promoting their conversion to T helper 17 (Th17) cells [123]. This may potentially cause P2X7 activation to be a “double edged sword” in GVHD progression, both impairing Tregs and promoting pathogenic Th17 cells. Consistent with this notion is that PPADS and/or BBG increase donor Treg survival in allogeneic [72] and humanised [117] mouse models of GVHD. Finally, studies of donor myeloid derived suppressor cells (MDSCs) in an allogeneic mouse model of GVHD provide indirect evidence for donor P2X7 in promoting this disease. Transplantation of donor MDSCs can impair GVHD development [124], however P2X7 activation of these cells prior to transplant limits their capacity to reduce GVHD in vivo [110]. Thus, this study suggests P2X7 on MDSCs present in the donor graft may impair the action of these suppressor cells to help promote GVHD in allo-HSCT recipients.

In conclusion, increases in extracellular ATP during GVHD lead to P2X7 activation on host APCs, which promotes cytokine release and donor T cell activation, with additional P2X7 activation on T cells directly promoting their proliferation and survival. Further, activation of P2X7 on donor Tregs and MDSCs may lead to reduced suppression of effector cells. Therefore, P2X7 remains a potential clinical target for treatment of GVHD.

## 8. P2Y_2_ and P2Y_12_ Receptor Signalling in GVHD

In addition to P2X7 a small number of studies implicate P2Y_2_, and to a lesser extent P2Y_12_, in the pathogenesis of GVHD [26,115]. P2Y_2_ expressing cells, mostly myeloid cells, are increased in the intestinal tract of allo-HSCT recipients with GVHD, with the presence of P2Y_2_-expressing cells associated with increased intestinal GVHD severity [125]. Notably, *P2ry2*^−/−^ recipient mice, but not *P2ry2*^−/−^ donor cells, displayed reduced clinical GVHD and increased survival, which corresponded to reduced histological GVHD, serum IL-6 concentrations and other inflammatory molecules compared to *wt* recipient mice [125]. Using bone marrow chimeras, a role of P2Y_2_ in GVHD was found to be more dependent on the host haematopoietic system than the non-haematopoietic system. The role of P2Y_2_ in the host haematopoietic system was due to its expression on inflammatory monocytes but not DCs [125]. Whilst the role of P2Y_2_ in the host non-haematopoietic system was hypothesised to be due to reduced P2Y_2_-mediated apoptosis of enterocytes (intestinal cells), with apoptosis of these cells reduced in *P2ry2*^−/−^ host mice compared to *wt* host mice following allo-HSCT [125].

P2Y_2_ may also contribute to GVHD progression by mediating the migration of DCs to sites of inflammation and ATP release [115], but as indicated above this is not an absolute requirement for GVHD development [125]. *mir-188*^−/−^ DCs, which cause less severe GVHD in an allogeneic mouse model, display reduced amounts of *P2ry2* and *P2ry12* compared to *wt* DCs [115]. Whilst *mir-188*^−/−^, *P2ry2*^−/−^ and *P2ry12*^−/−^ DCs all display impaired in vitro migration compared to *wt* DCs [115], evidence supporting a role for P2Y_2_ or P2Y_12_ in DC migration in GVHD in vivo is lacking.

Conversely, P2Y_2_ can limit GVHD progression by mediating migration of Tregs to sites of inflammation and ATP release. Co-injection of donor conventional T cells with donor *P2ry2*^−/−^ Tregs abrogates protection against GVHD afforded by the co-injection of donor conventional T cells with donor *wt* Tregs in an allogeneic mouse model [73]. Notably the addition of either donor *wt* or *P2ry2*^−/−^ Tregs reduced the expansion of donor conventional T cells to a similar extent in vivo, suggesting that the role of P2Y_2_ may be in directing Tregs to GVHD tissues to reduce inflammation at these sites. Thus, given the opposing roles of P2Y_2_ in DCs and Tregs in GVHD, systemic targeting of this receptor may be limited. To the best of our knowledge, no studies have reported the use of P2Y_2_ or P2Y_12_ antagonists to prevent GVHD in vivo.

A further role for P2Y_2_ in GVHD is in relation to dry eye disease, which is often associated with GVHD in allo-HSCT recipients [126]. P2Y_2_ activation is well known to induce the shedding of various cell surface molecules including tumour necrosis factor (TNF) receptor 1 (TNFR1) [127]. In the context of GVHD, soluble TNFR1 is increased in the tear fluid of patients compared to that of healthy controls, where it is thought to have a role in reducing inflammation on the eye surface by impairing TNFα-induced intracellular signalling [128]. Moreover, addition of the P2Y_2_ agonist diquafosol, an effective treatment of dry eye disease [129], increases the concentration of soluble TNFR1 in the tear fluid of people with GVHD [128]. Collectively, this suggests local activation of P2Y_2_ within the eye may limit the inflammation associated with dry eye disease in people with GVHD.

## 9. Adenosine Receptor Signalling in GVHD

As introduced above, ATP released at the sites of damage in allo-HSCT recipients promotes inflammation, but its degradation by apyrase can prevent GVHD [72] suggesting a beneficial role for the ecto-nucleotidases CD39 and CD73. In allogeneic mouse models of GVHD, pharmacological blockade of CD73 with APCP (days 0–6) [111] or APCP (twice weekly) [112] increased splenic CD4^+^ and CD8^+^ T cell numbers and serum IFN-γ and IL-6 concentrations and mouse mortality. As noted above, APCP also blocks CD39 [103], therefore a role for CD39 in this process cannot be excluded. However, genetic deficiency of CD73 in the donor or recipient also increased splenic CD4^+^ and CD8^+^ T cell numbers, serum IFNγ and IL-6 concentrations, histological damage in target organs, and mortality in allogeneic mouse models of GVHD [111,112] supporting a role for CD73 in limiting GVHD progression. Finally, a recent study demonstrated that murine CD150^hi^ Treg cells exhibit significantly higher *Entpd1* (CD39) and *Nt5e* (CD73) mRNA expression than CD150^-^ Tregs and mediate their suppressive effect through adenosine production, reducing histological GVHD and mortality, and promoting HSC expansion in an allogeneic mouse model of GVHD [130].

Further evidence for a role of CD39/CD73 in GVHD arises from studies using humanised mice. Transfer of human gingiva-derived mesenchymal stem cells into humanised NOD.Cg-*Prkdc^scid^* mice reduced histological damage, cytokine production by human T cells, and mouse mortality, however these effects were lost when these cells were incubated with the CD39 antagonist POM-1 prior to transplantation [116]. Blockade of CD39 and CD73 using the antagonist APCP (days 0–6), in a humanised NSG mouse model, increased weight loss, liver GVHD and serum human IL-2 concentrations [113]. Finally, mice injected with PBMCs from human donors with the intronic rs10748643 SNP in the *ENTPD1* gene (encoding CD39), which results in higher proportions of CD39^+^ Tregs and greater suppressive capacity [131], had worsened GVHD compared to NSG mice injected with PBMCs from donors without this SNP [132].

Adenosine produced by CD39/CD73 mediates its effects largely through activation of A_2A_ to limit GVHD progression. The role of A_2A_ in GVHD was originally shown in allogeneic mouse models of this disease (Table 3). The non-selective adenosine receptor antagonist caffeine (days 0–6, and 3–5 times a week onwards) worsened mortality [111]. Moreover, the A_2A_ antagonist SCH58261, but not the A_2B_ antagonist MRS1754 (days −2–12) increased serum TNFα, IFN-γ and IL-6, CD4^+^ and CD8^+^ T cell numbers, and worsened mortality [112]. Additionally, genetic deficiency of A_2A_ in the host increased CD4^+^ and CD8^+^ T cell numbers [111,112], but the effect on mortality was not reported in either study. Supporting the role of A_2A_ activation in limiting GVHD progression, treatment with the A_2A_ agonist ATL-146e increased serum IL-10, and reduced serum IFN-γ and IL-6 [133]. Furthermore, ATL-146e reduced activated splenic CD4^+^ and CD8^+^ T cell numbers, decreased T cell infiltration into GVHD tissue, and reduced both weight loss and mortality in an allogeneic mouse model of GVHD. ATL-146e, as well as the other A_2A_ agonists ATL-370 and ATL-1223, also increased donor derived Treg cells in both the skin and colon in this same model [134].

Further evidence for the role of A_2A_ in GVHD has been found using humanised mouse models. Human bone marrow derived mesenchymal stromal cells reduced IFN-γ- and TNF-α-producing leukocytes and reversed clinical disease in a humanised mouse model [135]. However, these beneficial effects were abrogated by the A_2A_ antagonist ZM241385 (from GVHD onset until endpoint), which prevented human T cell suppression, worsened disease, and reduced survival [135]. Contrary to allogeneic mouse models, injection of caffeine (days 0–14) did not impact weight loss, clinical score, survival, histology or serum cytokine concentrations in a humanised mouse model [113]. While injection of the A_2A_ agonist CGS21680 (days −2–11) had confounding effects in these mice, appearing to worsen disease by increasing weight loss and serum human IL-6 concentrations, but also potentially alleviating disease severity by reducing leukocyte infiltration into the liver and serum human TNFα concentrations [136].

## 10. Conclusions

The current review has briefly described how allo-HSCT is used in the treatment of haematological malignancies, reconstituting the immune system and providing a GVT effect. However, allo-HSCT also leads to GVHD, a severe immunological response against various tissues in the transplant recipient. Purinergic signalling is a modulator of HSC homeostasis, differentiation, mobilisation and death, and as such, plays a role in allo-HSCT. Furthermore, current evidence suggests that the purinergic system plays a variety of roles in GVHD progression (Figure 2) and as such, presents a viable target for developing new therapies. P2X7 and A_2A_ have pro- and anti-inflammatory effects, respectively, during GVHD, whilst CD39/CD73 exerts anti-inflammatory effects through the degradation of extracellular ATP and generation of extracellular adenosine. P2Y_2_ receptors play opposing roles on different cell types, and thus, may not be well-suited as a therapeutic target. Blockade of P2X7 or activation of A_2A_ present promising therapies for future GVHD treatment, but further studies in both allogeneic and humanised mouse models of this disease, as well as patients with GVHD, are required before these molecules can be targeted clinically.

## Figures and Tables

**Figure 1 ijms-22-08343-f001:**
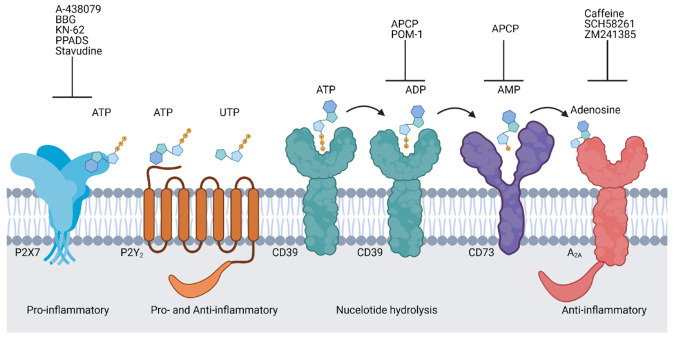
Purinergic signalling molecules associated with graft-versus-host disease. The purinergic receptors P2X7, P2Y_2_ and A_2A_ are activated by adenosine 5′-triphosphate (ATP), uridine triphosphate (UTP)/ATP or adenosine, respectively. P2X7 plays a pro-inflammatory role and A_2A_ plays an anti-inflammatory role while P2Y_2_ may be involved in both roles. CD39 converts ATP to adenosine 5′-diphosphate (ADP) and ADP to adenosine 5′-monophosphate (AMP). CD73 converts AMP to adenosine. Antagonists for P2X7 (A-438079, Brilliant blue G (BBG), KN-62, and pyridoxalphosphate-6-azophenyl-2’,4’-disulfonic acid (PPADS) and stavudine), CD39 (α,β-methylene ADP (APCP) and polyoxotungstate-1 (POM-1)), CD73 (APCP) and A_2A_ (Caffeine, SCH58261 and ZM241385) have been used to study the roles of these molecules in GVHD. Created with BioRender.com.

**Figure 2 ijms-22-08343-f002:**
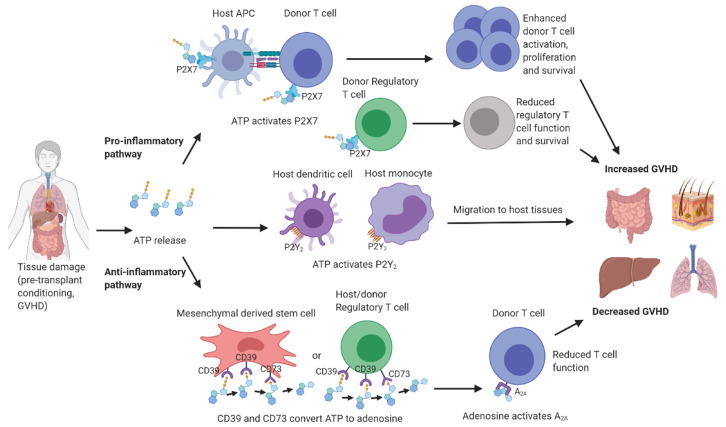
The role of the purinergic signalling molecules during graft-versus-host disease. Tissue damage from pre-transplant conditioning or GVHD results in adenosine 5′-triphosphate (ATP) release. This ATP activates P2X7 on host antigen presenting cells (APCs) and donor T cells leading to donor T cell enhanced activation, proliferation and survival. ATP can also activate P2X7 on regulatory T cells, reducing their function and survival. P2X7 activation on myeloid derived suppressor cells can reduce their suppressive capacity (not shown). Combined these pathways result in worsened GVHD. ATP can also activate P2Y_2_ on host dendritic cells and monocytes initiating the migration of these cells to inflamed tissues. Alternatively, ATP can be converted to ADP then AMP by CD39 present on mesenchymal stem cells or regulatory T cells. CD73 receptors on these same cells, or other cells (not shown) convert AMP to adenosine which activates A_2A_ receptors on donor T cells, reducing T cell function. This conversion of ATP to adenosine results in reduced GVHD. Created with BioRender.com.

**Table 1 ijms-22-08343-t001:** Extracellular ATP degradation and blockade of CD39 and CD73 in mouse models of graft-versus-host disease.

Model	Graft	Host	Target	Drug Regime	Outcomes	Ref.
Allogeneic	5 × 10^6^ TCD ^a^ BM ^b^ cells + 1.6 × 10^6^ (FVB/N) or 1 × 10^6^ CD4^+^ /CD8^+^ (C57BL/6) (i.v. ^c^)	BALB/c	eATP ^d^	4U Apyrase (hydrolysis catalyst) i.p. ^e^ days 0–2, 6–8	↑^f^ Survival ↓^g^ Apoptosis and inflammation in the gastrointestinal tract ↓ Serum IFN ^h^-γ	[72]
Allogeneic	10 × 10^6^ CD25 depleted T cells (C57BL/6) (injection route not disclosed)	BALB/c	eATP	4U Apyrase (hydrolysis catalyst) i.p. days 0–4	↑ Survival	[110]
Allogeneic	5 × 10^6^ BM cells and 2 × 10^5^ CD4^+^/CD8^+^ splenic T cells (i.v.)	C57BL/6	CD73	50 mg/kg APCP ^i^ (antagonist) i.p. days 0–6	↑ Mortality	[111]
Allogeneic	5 × 10^6^ BALB/c splenocytes ± BM cells or splenocytes (i.v.)	C57BL/6	CD73	20 mg/kg APCP (antagonist) i.v. twice weekly	↑ Mortality ↑ Splenic CD4^+^ and CD8^+^ T cells ↑ Serum IFN-γ and IL^j^-6.	[112]
Humanised	10 × 10^6^ hPBMCs ^k^ (i.p.)	NSG ^l^	CD39 and CD73	50 mg/kg APCP (antagonist) i.p. days 0–6	↑ Weight loss ↑ Histological damage ↑ Serum human IL-2	[113]
Humanised	20 × 10^6^ CD25-depleted hPBMC + PBS, GMSCs ^m^, human fibroblasts or nTregs ^n^ (i.v.)	NOD.Cg-*Prkdc^scid^*	CD39	100 μM POM-1 ^o^ (antagonist) was used to pretreat GMSCs	↑ Mortality ↑ Histological damage ↑ Serum IL-4, IL-17, IFN-γ, IL-2 and TNF^p^-α.	[116]

^a^ T cell depleted, ^b^ bone marrow, ^c^ intravenous, ^d^ extracellular adenosine 5′- triphosphate, ^e^ intraperitoneal, ^f^ increased, ^g^ decreased, ^h^ interferon, ^i^ α,β-methylene adenosine diphosphate, ^j^ interleukin, ^k^ human peripheral blood mononuclear cells, ^l^ NOD.Cg-*Prkdc^scid^Il2rg^null^*, ^m^ gingival mesenchymal stem cells, ^n^ natural regulatory T cells, ^o^ polyoxotungstate-1, ^p^ tumour necrosis factor.

**Table 2 ijms-22-08343-t002:** P2X7 blockade in mouse models of graft-versus-host disease.

Model	Graft	Host	Target	Drug Regime (All Antagonists)	Outcomes	Ref.
Allogeneic	5 × 10^6^ TCD ^a^ BM ^b^ cells + 1.6 × 10^6^ (FVB/N) or 1 × 10^6^ CD4^+^ /CD8^+^ (C57BL/6) (i.v. ^c^)	BALB/c	P2X7	10 µmol PPADS ^d^ i.p. ^e^ days 0–10	↑^f^ Survival ↑ Tregs^g^, ↓ T cell expansion ↓^h^ Serum IFN^i^-γ ↓ GVHD severity	[72]
Allogeneic	5 × 10^6^ TCD BM cells + 1 × 10^6^ CD4^+^ /CD8^+^ (C57BL/6) (i.v.)	BALB/c	P2X7	1 µmol KN62 i.p. days 0–10	↑ Survival	[72]
Allogeneic	10 × 10^6^ CD25 depleted T cells (C57BL/6) (injection route not disclosed)	BALB/c	P2X7	80 mg/kg A-438079 i.p. day 0–4	↑ Survival	[110]
Allogeneic	10 × 10^6^ TCD BM cells + 2.5 × 10^6^ CD4^+^ T cells (C57BL/6) (injection route not disclosed)	BALB/c	P2X7	25 mg/kg Stavudine (d4T) i.p. twice daily from day −1 or 0	↑ Survival (when started from day −1) ↓ Serum IFN-γ, TNF^j^-α and IL^k^-6 ↓ Liver inflammation	[100]
Allogeneic	5 × 10^6^ BM cells + 5 × 10^6^ splenic cells (C57BL/6) (i.v.)	BALB/c	P2X7	50 mg/kg or 75mg/kg BBG^l^ i.p. twice weekly for 4 weeks	↓ Weight loss ↓ Liver inflammation ↓ CXCL8 and CCL2, *Il1B* and *Il18*	[114]
Humanised	10 × 10^6^ hPBMCs^m^ (i.p.)	NSG^n^	P2X7	50 mg/kg BBG i.p. days 0, 2, 4, 6, 8, 10	↓ Serum IFN-γ ↓ Liver, skin and small intestine inflammation	[118]
Humanised	10 × 10^6^ hPBMCs (i.p.)	NSG	P2X7	50 mg/kg BBG i.p. thrice weekly until endpoint	↓ Liver inflammation	[119]
Humanised	10 × 10^6^ hPBMCs (i.p.)	NSG	P2X7	50 mg/kg BBG i.p. days 0–10	↓ Clinical disease ↓ Liver and skin inflammation ↓ Serum IFN-γ ↑ Tregs and B cells	[117]
Humanised	10 × 10^6^ hPBMCs (i.p.)	NSG	P2X7	300 mg/kg PPADS i.p. days 0–10	↑ Tregs	[117]

^a^ T cell depleted, ^b^ bone marrow, ^c^ intravenous, ^d^ pyridoxalphosphate-6-azophenyl-2’,4’-disulfonic acid, ^e^ intraperitoneal, ^f^ increased, ^g^ regulatory T cells, ^h^ decreased, ^i^ interferon, ^j^ tumour necrosis factor, ^k^ interleukin, ^l^ Brilliant blue G, ^m^ human peripheral blood mononuclear cells, ^n^ NOD.Cg-*Prkdc^scid^Il2rg^null^*.

**Table 3 ijms-22-08343-t003:** Adenosine receptor blockade or activation in mouse models of GVHD.

Model	Graft	Host	Target	Drug Regime	Outcomes	Ref.
Allogeneic	5 × 10^6^ BM^a^ cells and 2 × 10^5^ CD4^+^/CD8^+^ splenic T cells (i.v.^b^)	C57BL/6	Adenosine receptors	10 mg/kg Caffeine (antagonist) i.p.^c^ days 0–6 then 3–5 times a week	↑^d^ Mortality	[111]
Allogeneic	5 × 10^6^ BALB/c splenocytes ± BM cells or splenocytes (i.v.)	C57BL/6	A_2A_	2 mg/kg SCH58261 (antagonist) i.p. days −2–12	↑ Mortality ↑ Splenic CD4^+^ and CD8^+^ T cells ↑ Serum IFN^e^-γ, TNF^f^-α and IL^g^-6	[112]
Allogeneic	5 × 10^6^ BALB/c splenocytes ± BM cells or splenocytes (i.v.)	C57BL/6	A_2B_	2 mg/kg MRS1754 (antagonist) i.p. days −2–12	No differences	[112]
Allogeneic	10 × 10^6^ C57BL/6 splenocytes ± 10 × 10^6^ T cells (i.v.)	B6D2F1/J	A_2A_	10 ng/kg ATL-146e (agonist) s.c. days 0–14	↑ Survival ↑ Serum IL-10 ↓^h^ Splenic CD4^+^ and CD8^+^ T cells ↓ T cell infiltration into target organs ↓ Serum IFN-γ and IL-6	[133]
Humanised	5 × 10^6^ human Th1^i^ cells and 3 × 10^6^ human monocytes (i.p.)	NSG^j^	A_2A_	1.5 mg/kg ZM241385 (antagonist) i.p. daily from disease onset until endpoint	↑ Mortality ↑ Histological damage ↑ IFN-γ and TNF-α producing cells	[135]
Humanised	10 × 10^6^ hPBMCs^k^ (i.p.)	NSG	Adenosine receptors	10 mg/kg Caffeine (antagonist) i.p. days 0–14	No differences	[113]
Humanised	10 × 10^6^ hPBMCs (i.p.)	NSG	A_2A_	0.1 mg/kg CGS21680 (agonist) i.p. day −2–11	↑ Weight loss ↑ Serum human IL-6 ↓ Histological damage ↓ Serum human TNF-α	[136]

^a^ Bone marrow, ^b^ intravenous, ^c^ intraperitoneal, ^d^ increased, ^e^ interferon, ^f^ tumour necrosis factor, ^g^ interleukin, ^h^ decreased, ^i^ T helper 1, ^j^ NOD.Cg-*Prkdc^scid^Il2rg^null^*, ^k^ human peripheral blood mononuclear cell.

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
