# Peer review of "Purinergic Signalling in Allogeneic Haematopoietic Stem Cell Transplantation and Graft-versus-Host Disease"

_ijms, 2021, doi:10.3390/ijms22158343_

Round 1
Reviewer 1 Report
The presented review deals with a very interesting topic that is still relevant.
The review is written in a very readable way, it contains a summary of a large amount of experimental data, which provides a comprehensive overview of the issue.
I have only a very small comments on the text. Chapter 2 (Hematopoietic malignancies) is unnecessary in the review, the types of malignancies are not discussed in the text, so there is no reason to give an introduction to the types of malignancies in the review. I would recommend deleting this chapter altogether. Moreover, please, use only "haematological" instead of "haematopoietic" malignancies (unify within the text)
Line 250 - due to irreversible infection-related deaths. I recommend rewriting, this is how death seems to be irreversible with the possibility of reversible death (only due to infection-related deaths are enough).
Author Response
We thank both reviewers for their insightful and helpful comments.
The presented review deals with a very interesting topic that is still relevant. The review is written in a very readable way, it contains a summary of a large amount of experimental data, which provides a comprehensive overview of the issue.
1) I have only a very small comments on the text. Chapter 2 (Hematopoietic malignancies) is unnecessary in the review, the types of malignancies are not discussed in the text, so there is no reason to give an introduction to the types of malignancies in the review. I would recommend deleting this chapter altogether.
Response: Section 2 (Hematopoeitic malignancies) has been removed from the text. The first sentence from this section has been rephrased and added to the beginning of Section 1 to provide context (lines 38-40).
‘Haematopoietic malignancies’ has been removed from the final paragraph in Section 1 as the review no longer provides a brief overview of this topic. NB: This change has also led to the renumbering of subsequent sections.
2) Moreover, please, use only "haematological" instead of "haematopoietic" malignancies (unify within the text).
Response: With the removal of Section 2 ‘Haematopoeitic Malignancies’ the review uses the term‘haematological malignancies’ instead of ‘haematopoeitc malignancies’ throughout as requested.
3) Line 250 - due to irreversible infection-related deaths. I recommend rewriting, this is how death seems to be irreversible with the possibility of reversible death (only due to infection-related deaths are enough).
Response: The word ‘irreversible’ has been removed from this sentence for clarity (line 253).
Reviewer 2 Report
The text is very interesting because it describes in a very interesting way the diagnostic and possibly therapeutic perspectives of some nucleoside mediators and their receptors in the context of the prevention and treatment of GvHD post allogeneic stem cell transplantation. The tables are appropriate and illustrative of the studies conducted with agonists and antagonists in the mouse and humanized models.
The section of results and conclusions are congruent with the proposed results.
Comments:
1) The introductory section concerning transplantation and that relating to haematological diseases can be reduced to a single paragraph as it is not the target of the study.
2) It would be important for the authors to correlate the results of this review with the drugs that are routinely used for the treatment of GvHD. How do prednisone or other corticosteroid drugs interface with purinergic mediators? This relationship could be of relevant clinical significance since prednisone, in addition to having a direct lymphocytolytic effect, also acts on other cellular stems such as neutrophils and macrophages. The failure of steroid therapy that appears in about 50% of patients with acute GvHD could be linked to new mechanisms such as those described in the paper.
Author Response
We thank both reviewers for their insightful and helpful comments.
The text is very interesting because it describes in a very interesting way the diagnostic and possibly therapeutic perspectives of some nucleoside mediators and their receptors in the context of the prevention and treatment of GvHD post allogeneic stem cell transplantation. The tables are appropriate and illustrative of the studies conducted with agonists and antagonists in the mouse and humanized models.
The section of results and conclusions are congruent with the proposed results.
Comments:
1) The introductory section concerning transplantation and that relating to haematological diseases can be reduced to a single paragraph as it is not the target of the study.
Response: Please see response to Comment 1 from Reviewer 1, which has been copied here for Reviewer 2's convenience as follows: Section 2 (Hematopoeitic malignancies) has been removed from the text. The first sentence from this section has been rephrased and added to the beginning of Section 1 to provide context (lines 38-40). ‘Haematopoietic malignancies’ has been removed from the final paragraph in Section 1 as the review no longer provides a brief overview of this topic. NB: This change has also led to the renumbering of subsequent sections.
2) It would be important for the authors to correlate the results of this review with the drugs that are routinely used for the treatment of GvHD. How do prednisone or other corticosteroid drugs interface with purinergic mediators? This relationship could be of relevant clinical significance since prednisone, in addition to having a direct lymphocytolytic effect, also acts on other cellular stems such as neutrophils and macrophages. The failure of steroid therapy that appears in about 50% of patients with acute GvHD could be linked to new mechanisms such as those described in the paper.
Response: The interaction between steroids and purinergic pathways is largely unknown. We have added a paragraph (lines 363-372) at the end of Section 6 ‘Purinergic signalling in GVHD’ discussing how corticosteroids and other GVHD treatments may interact with purinergic pathways.